# The Role of PHLDA3 in Cancer Progression and Its Potential as a Therapeutic Target

**DOI:** 10.3390/cancers17071069

**Published:** 2025-03-22

**Authors:** Walied A. Kamel, Jayaraman Krishnaraj, Rieko Ohki

**Affiliations:** 1Laboratory of Fundamental Oncology, National Cancer Center Research Institute, Tsukiji 5-1-1, Chuo-ku, Tokyo 104-0045, Japan; awari-do@ncc.go.jp (W.A.K.); kjayaram@ncc.go.jp (J.K.); 2Department of Zoology, School of Science, Mansoura University, Mansoura 35516, Egypt

**Keywords:** PHLDA3, p53, AKT pathway, tumor progression, metastasis, invasion

## Abstract

PHLDA3 acts as a key inhibitor of the epithelial–mesenchymal transition (EMT) and tumor progression, including metastasis, by modulating the Phosphoinositide 3-kinases/Protein kinase B (PI3K/AKT) signaling pathway. Downregulation of PHLDA3 is associated with increased mesenchymal EMT marker expression, higher metastatic potential, and accelerated cancer progression in squamous cell carcinomas (SCC). *PHLDA3* mRNA levels have been shown to correlate with prognosis in various cancers, including SCC and neuroendocrine tumors, where reduced levels indicate poorer outcomes. This makes PHLDA3 a valuable prognostic marker, and its role in suppressing cancer progression highlights its potential as a promising therapeutic target.

## 1. Introduction

Cancer is a group of diseases characterized by uncontrolled cell growth and metastasis, which can be life-threatening. Its development is influenced by external factors such as tobacco, chemicals, radiation, and infections, as well as internal factors such as genetic mutations, hormones, and immune conditions. While the exact causes remain complex, risk factors like diet, obesity, inactivity, and environmental pollutants contribute to carcinogenesis, making cancer a leading cause of death [1]. Currently, the United States is expected to experience a staggering 2 million new cancer diagnoses annually, which equates to approximately 5480 daily cases [2]. In 2020, an estimated 18.1 million new cancer cases (excluding non-melanoma skin cancer) were diagnosed worldwide [3]. This alarming statistic highlights the need to investigate the biological processes that drive cancer, including various cellular regulatory pathways, such as the Protein Kinase B (AKT) and Wnt signaling pathways.

We and others have reported that the PHLDA3 protein, which belongs to the pleckstrin homology-like domain family A, inhibits the AKT pathway and suppresses tumors [4,5,6]. Both PHLDA3 and AKT protein possess pleckstrin homology-like (PHL) domains that exhibit high affinity for phosphatidylinositol phosphates (PIPs) located in the inner leaflet of the cell membrane [7]. As a result, PHLDA3 binds competitively and with a higher affinity to PIPs than AKT, thereby impeding the vital membrane translocation and activation of AKT [4]. Therefore, PHLDA3 protein significantly contributes to cellular signaling and tumor suppression as a consequence of its specific membrane lipid binding through its PH domain [5,7,8]. Furthermore, other studies have documented that *PHLDA3* is one of the genes directly stimulated by p53, resulting in increased PHLDA3 protein production, which suppresses cell proliferation and promotes cell death [5,9,10].

Moreover, many studies have reported the tumor-suppressive effects of PHLDA3 in various cancer types [4,5,10,11,12,13]. These studies consistently report decreased expression of *PHLDA3* in cancers such as neuroendocrine tumors (NETs) [5,12], squamous cell carcinoma (SCC) [10,13], esophageal SCC (ESCC) [13], and prostate cancer [4]. These observations indicate that decreased levels of PHLDA3 protein are linked to cancer, providing evidence for its tumor-suppressive role. Given the increasing prevalence of cancer worldwide, understanding the underlying mechanisms of the disease, including the potential role of PHLDA3 as a tumor-suppressor gene, is critically important. This review provides a comprehensive understanding of PHLDA3’s complex involvement in cancer progression, including metastasis and invasion, highlighting its therapeutic implications for advanced-stage malignancies. Insights gained from an understanding of the role of PHLDA3 could pave the way for developing novel treatments and improving therapeutic strategies against these aggressive cancers.

## 2. Regulation of *PHLDA3* Transcription by p53

p53 has been called the “guardian of the genome” and protects genetic material by reacting to cellular stresses such as DNA damage, oxygen deprivation, and oncogene activation [14,15]. These stresses trigger p53 activation by phosphorylation and acetylation, leading to its stabilization and relocation to the cell nucleus [16]. p53 acts as a transcription factor within the nucleus, activating genes like *PHLDA3* [5]. p53 regulates cellular responses by activating genes that repair or eliminate damaged cells by regulating apoptosis, cell cycle arrest, autophagy, and senescence to prevent tumor formation [17]. A p53-binding site in the promoter region of *PHLDA3* was predicted by computational analysis. This prediction was validated by ChIP-chip, ChIP-sequencing, and Global Run-On sequencing experiments, which definitively demonstrated p53’s direct binding to this specific DNA sequence [18,19]. This confirms *PHLDA3* as a bona fide target gene regulated by the p53 tumor suppressor pathway (Figure 1).

## 3. PHLDA3 Is a Repressor of AKT Oncoprotein

PHLDA3 (Pleckstrin Homology Like Domain Family A Member 3) is a gene that encodes a protein consisting of 127 amino acids, most of which is a PH domain (amino acids 7-108) (Figure 1A). PHLDA3 binds to various phosphatidylinositol phosphates (PIPs) through this domain and thereby localizes itself to the cell membrane [5]. AKT, a well-established oncogene, frequently exhibits abnormal tumor activation due to various extracellular stimuli such as growth factors and hormones [20,21,22,23]. Upstream of AKT is phosphoinositide 3-kinase (PI3K), a lipid kinase that promotes cell proliferation and cancer development by converting phosphatidylinositol 4,5-bisphosphate (PIP2) to phosphatidylinositol (3,4,5)-trisphosphate (PIP3) [23,24]. Activation of AKT depends on its ability to bind to PIP3, facilitated by a PH domain within its N-terminus [25]. When PIP3 accumulates in the membrane, AKT is relocated to the membrane and binds to PIP3, triggering AKT phosphorylation and subsequent activation. This activated AKT, a serine/threonine kinase, governs cell proliferation by phosphorylating downstream targets such as mouse double-minute 2 homolog (MDM2), glycogen synthase kinase 3 (GSK3), tuberous sclerosis complex 2, and mammalian target of rapamycin complex 1 (mTORC1). This intricate phosphorylation network plays a central role in cell survival signaling [25,26]. *PHLDA3* acts as a dominant-negative repressor of AKT signaling. It inhibits AKT activation and its downstream pro-survival signaling by directly competing with AKT for PIP binding on the plasma membrane [5] (Figure 1B).

## 4. Dual Roles of PHLDA3 in the WNT Signaling Pathway

The Wnt signaling pathway is a conserved cell communication system that regulates embryonic development, tissue homeostasis, and can influence the progression of diseases such as cancer. It can be divided into canonical (β-catenin-dependent) and non-canonical (β-catenin-independent) pathways that regulate cell proliferation, differentiation, migration, and apoptosis. In the canonical pathway, Wnt ligand binding inhibits GSK3β, allowing β-catenin accumulation and activation of Wnt target genes that drive tumor growth. The non-canonical pathway regulates cell polarity and calcium signaling. Dysregulated Wnt signaling is linked to cancer, developmental disorders, and stem cell regulation, making it a key therapeutic target for disease treatment and regenerative medicine [27,28,29]. Furthermore, studies by Schneider et al. and Yeh et al. found that Wnt/β-catenin pathway hyperactivation is a critical driver of prostate cancer progression and resistance to therapy [30,31]. Ma et al. [4] reported that PHLDA3 protein overexpression in prostate cancer suppresses β-catenin activity by decreasing phosphorylated GSK3β, promoting β-catenin degradation and downregulating Wnt target gene expression.

A study by Chen et al. [13] on ESCC found that *PHLDA3* and BarH-like homeobox 2 (BARX2), a downstream target of Wnt/β-catenin signaling, inhibit gastric cancer progression. This study demonstrated that BARX2 indirectly activates *PHLDA3* transcription, leading to PI3K/AKT suppression and subsequent inhibition of ESCC cell proliferation, migration, invasion, and angiogenesis. Notably, knockdown of either *BARX2* or *PHLDA3* abrogated these suppressive effects, highlighting their role in suppressing tumors [13].

Conversely, a study by Lei et al. [32] reported that PHLDA3 protein overexpression promotes lung adenocarcinoma cell proliferation and invasion by activating the Wnt signaling pathway and EMT. This effect is mediated by PHLDA3 protein binding to and inactivating GSK3β, a negative regulator of Wnt signaling. PHLDA3 protein augmented Wnt signaling by increasing the expression and activation of β-catenin, as well as its downstream targets MYC Proto-Oncogene, BHLH Transcription Factor (Myc), cyclin D1, and matrix metallopeptidase 7 (MMP7). The proliferative and invasive abilities induced by PHLDA3 protein overexpression were eliminated by inhibition of Wnt signaling with XAV-939 or GSK3β activity with CHIR-99021.

Collectively, these results illustrate the multifaceted roles of PHLDA3 protein in modulating the Wnt signaling pathway across different types of cancer. Interestingly, PHLDA3 does not always act as a tumor suppressor, as its function can vary depending on context and its interactions with other signaling pathways. Future studies should explore how PHLDA3 exerts its oncogenic effects in specific cancer subtypes and whether it can be selectively targeted to suppress tumor growth without promoting malignancy in other contexts (Figure 2).

## 5. PHLDA3 in Apoptosis

The inhibitory effect of PHLDA3 on AKT complements the pro-apoptotic role of the p53 tumor suppressor protein [5]. Furthermore, overexpression of PHLDA3 protein has been demonstrated to directly induce caspase activation, a hallmark of the execution phase of apoptosis [19,33]. These findings demonstrate that PHLDA3 protein is a critical mediator of apoptosis that functions by inhibiting pro-survival signaling and cooperating with the p53 pathway (Figure 3A). Other studies have demonstrated that PHLDA3 can activate a pro-apoptotic pathway in response to hypoxia, highlighting a potential mechanism by which p53 can mediate cell death under hypoxic stress [34,35].

A study by Bensellam et al. [36] revealed that *PHLDA3* plays a critical role in maintaining β cell function and survival in the context of both type 1 and type 2 diabetes, where β cells face multiple stresses such as cytokine-mediated inflammation, oxidative stress, and endoplasmic reticulum (ER) stress. This study utilized a pancreatic β cell model, investigating PHLDA3 expression in both in vivo and in vitro systems. In vivo, the researchers analyzed pancreatic islets from diabetic humans and mouse models, including db/db mice (a type 2 diabetes model) and non-obese diabetic (NOD) mice (a type 1 diabetes model). In vitro, they exposed isolated pancreatic islets and MIN6 β cells to various stressors, such as inflammatory cytokines and palmitate, as well as oxidative stress-inducing agents like ribose and hydrogen peroxide. Their findings demonstrated that PHLDA3 expression is significantly upregulated under these conditions, particularly through the unfolded protein response (UPR), a key adaptive mechanism in β cells facing metabolic and inflammatory stress. Research has shown that *PHLDA3* expression is significantly upregulated in response to these stress conditions in both in vivo and in vitro models. The transcription of PHLDA3 is indirectly controlled by X-box binding protein 1 (XBP1), which drives its expression. This defines PHLDA3 as a key mediator in balancing β cell survival and stress responses. Functionally, PHLDA3 represses expression of the pro-inflammatory inducible nitric oxide synthase gene (iNOS), which is induced by nuclear factor κB (NFκB) and supports the expression of the antioxidant genes (Glutathione peroxidase 1 (Gpx1) and sulfiredoxin 1 homolog (Srxn1)). Knockdown experiments have demonstrated that the absence of PHLDA3 exacerbates cytokine-induced apoptosis and inflammation while weakening antioxidant defenses, underscoring its protective role in β cells [36]. We also found that a deficiency in PHLDA3 protein results in increased islet proliferation and inhibition of apoptosis [35]. Collectively, these data strongly suggest that the PHLDA3 protein acts as a robust defender of β cell health under various stress conditions. It appears to exert its protective effects by quelling inflammation and bolstering antioxidants and adaptive UPR responses (Figure 3B).

## 6. *PHLDA3* in Irradiation

Multiple studies have highlighted the critical role of p53 in regulating the cellular response to radiation [37,38]. *PHLDA3* has emerged as a key component of a radiation-responsive gene signature, highlighting its potential as a valuable biomarker for radiation exposure assessment. Traditional methods of studying this response, such as ex vivo irradiation of human blood, face significant limitations, including the lack of tissue-specific signaling and the progressive degradation of blood cells in culture. However, a recent novel approach using in vivo mouse models addresses these challenges by providing a more comprehensive understanding of gene expression in irradiated blood cells [39]. Notably, the inclusion of *PHLDA3* in the gene panel enhances the sensitivity and reliability of radiation dose reconstruction. Importantly, the findings demonstrated minimal sex-specific differences in gene expression, underscoring the robustness of *PHLDA3* as part of a high-throughput solution for assessing radiation exposure in real-world scenarios [39].

Hasapis et al. [40] reported that ionizing radiation increases the expression of PHLDA3 protein in hematopoietic cells, mediated by the p53 tumor suppressor pathway. Using isogenic cell lines and genetically modified mouse models, they demonstrated that ionizing radiation triggers *PHLDA3* upregulation in both human leukemia cells and normal murine hematopoietic cells. Interestingly, despite the induction of PHLDA3 protein expression by radiation, deletion of the *PHLDA3* gene did not alleviate the acute hematopoietic toxicity caused by radiation exposure or protect against radiation injury or the development of lymphoma. However, deletion of *PHLDA3* did decrease the number of genetic mutations required for radiation-induced lymphomagenesis. This suggests that PHLDA3 protein may have an impact on genomic stability following radiation exposure. Additionally, altered PHLDA3 expression in lymphomas suggests its potential role in post-radiation malignancies [40].

Furthermore, Furukawa et al. [41] investigated p53–PHLDA3 signaling as a potential therapeutic target for cardiac hypertrophy. They used a transverse aortic constriction (TAC) to induce cardiac hypertrophy in mice, followed by X-ray (20 or 30 Gy) or carbon-ion beam (10 or 30 Gy) irradiation. They found that TAC treatment inhibited the expression of PHLDA3 protein. Similarly, we also found that *PHLDA3* expression is responsive to UV and gamma radiation exposure [19,33]. These findings indicate that *PHLDA3* may be crucial in the cellular response to radiation, offering a potential target for therapies aimed at reducing radiation-induced damage and preventing post-radiation malignancies.

## 7. *PHLDA3* in ESCC, Osteosarcoma, Acute Myeloid Leukemia, B-Cell Lymphoma, and Prostate Cancer Cell Lines

Tumor-suppressive roles of PHLDA3 protein have been reported in several cancer cell lines including ESCC, osteosarcoma (OS), acute myeloid leukemia (AML), B-cell lymphoma, and prostate cancer [6,13,42]. *PHLDA3* was found to be downregulated in ESCC cells. PHLDA3 upregulation has the functional effect of suppressing the PI3K/AKT signaling pathway, which leads to the inhibition of ESCC cell proliferation, migration, invasion, and angiogenesis [13]. PHLDA3 activation also plays a functional role in reducing OS cell proliferation, migration, and chemoresistance. Conversely, inhibiting PHLDA3 had opposing effects [6]. Furthermore, miR-19a-3p has been identified as a potential oncogenic driver that inhibits *PHLDA3* expression in OS. The results confirm that the miR-19a-3p/*PHLDA3* pathway plays a crucial role in OS and emphasizes the potential of PHLDA3 as a therapeutic target [6].

In AML, the PHLDA3 protein plays a diverse role in regulating cell survival and apoptosis. PHLDA3 interacts with other proteins, such as CREB-binding protein/E1A-associated protein (Cbp/p300)-interacting transactivator with Glu/Asp-rich carboxy-terminal domain 2 (CITED2), to regulate these processes. CITED2 knockdown increased *PHLDA3* expression, which triggered apoptosis in AML cells. However, the simultaneous inhibition of both CITED2 and PHLDA3 resulted in a partial rescue of cells from apoptosis, thus emphasizing the potential role of PHLDA3 protein in cell survival mechanisms [42].

PHLDA3 significantly impacts the B-cell receptor (BCR) signaling pathway in the context of B-cell lymphoma. By blocking AKT signaling, a key component of the BCR signaling pathway, PHLDA3 suppresses the growth and survival signals transmitted through the BCR pathway. The activation of PHLDA3, which has been observed during protein arginine methyltransferase 5 inhibition in mantle cell lymphoma, helps to suppress BCR signaling and its downstream effects. This suppression impacts the proliferation and survival of B-cell lymphoma cells [43].

Moreover, a study reported that in prostate cancer cell lines, PHLDA3 levels are low. The promoter region of *PHLDA3* is highly methylated in prostate cancer. Treatments that effectively reverse methylation resulted in a significant increase in PHLDA3 protein levels. Overexpression of PHLDA3 in these cells resulted in a deceleration of their growth, a reduction in their division rate, and inhibition of the EMT process, all of which play crucial roles in tumor progression [4].

In summary, PHLDA3 plays a significant role in suppressing tumors across various cancer types, impacting key signaling pathways and cellular processes involved in cancer progression. This highlights PHLDA3 as a promising therapeutic target for cancer treatment as shown in Table 1.

## 8. PHLDA3 in SCC: Functional Roles in Tumor Progression and Data Therapeutic Resistance

### 8.1. Cutaneous SCC (cSCC)

Analysis of the role of PHLDA3 in cSCC has revealed its significant impact on tumor progression, including metastasis. *PHLDA3* is expressed in normal interfollicular epidermis and in hair follicles. In hair follicle stem cells, the origin of metastatic cSCC, the *PHLDA3* genomic locus shows active chromatin marks. A mouse model of SCC that accurately mimics the progression of the disease in humans is induced by DMBA (7,12-dimethylbenz[a]anthracene) and TPA (12-O-tetradecanoylphorbol-13-acetate). In this model, it was shown that while PHLDA3 does not influence the papilloma to carcinoma transition, the loss of the p53–PHLDA3 pathway is linked to enhanced metastasis, underscoring PHLDA3’s role in suppressing metastatic tumor formation. Furthermore, the loss of PHLDA3 protein complements p53 mutations, emphasizing its importance in the progression of SCC. Mechanistically, decreased PHLDA3 expression promotes EMT in SCCs, contributing to their metastatic potential. This is evidenced by elevated EMT markers such as vimentin and N-cadherin in PHLDA3-deficient SCCs (Figure 4) [10]. This study extends beyond the mouse model, revealing a correlation between low PHLDA3 expression and poor prognosis in human SCC patients. These findings indicate that PHLDA3 is a key downstream effector of p53 signaling, acting as a potent suppressor of SCC development and metastasis [10].

### 8.2. ESCC and Head and Neck SCC (HNSCC) 

PHLDA3 protein was found to be downregulated in ESCC tissues and cells. Its upregulation led to the suppression of the PI3K/AKT signaling pathway, resulting in reduced ESCC cell proliferation, migration, invasion, and angiogenesis [13]. Previous studies have also found that the expression of PHLDA3 is reduced in ESCC, which has been associated with patient survival and prognosis. Furthermore, low PHLDA3 expression correlates with poor prognosis in ESCC patients, including increased risk of postoperative tumor progression and recurrence [44]. The current understanding of the role of PHLDA3 protein in the pathogenesis of HNSCC is that it acts as a negative feedback regulator of the PI3K pathway. In HNSCC, the expression levels of PHLDA3 have been found to be upregulated, indicating its potential involvement in the dysregulation of the PI3K pathway. This dysregulation of the PI3K pathway is believed to play a significant role in the development and progression of HNSCC [11].

## 9. The Role of PHLDA3 in Neuroendocrine Tumors

### 9.1. Function of PHLDA3 in Islet Β Cells

To investigate the role of PHLDA3 in islet β cell hyperplasia and resistance to apoptosis, we conducted an in vivo study using *PHLDA3*-deficient mice. Our experiments revealed several key findings. First, we observed increased phosphorylation of AKT and its downstream substrates, such as Ribosomal protein S6 kinase beta-1 (p70S6K), Ribosomal protein S6 (S6), GSK-3β, and MDM2, indicating abnormal activation of the AKT pathway in these mice. This heightened AKT activity led to a significant proliferation of islet β cells, resulting in hyperplasia. The activation of the AKT pathway is known to enhance cell survival by inhibiting apoptotic processes. Consistent with this, we found that *PHLDA3*-deficient mice exhibited resistance to apoptosis in their islet β cells. The dual effect of promoting cell proliferation and preventing apoptosis contributed to the marked expansion of islet β cells and the development of hyperplasia. Additionally, we observed altered glucose metabolism in *PHLDA3*-deficient mice, with elevated insulin levels and lower blood glucose concentrations under fed conditions. Glucose tolerance tests further confirmed enhanced glucose tolerance in these mice. Interestingly, the data also revealed haplo-insufficiency of *PHLDA3* for suppressing islet cell proliferation, as both heterozygous and knockout mice displayed islet hyperplasia. These findings demonstrate the critical function of PHLDA3 in regulating islet cell proliferation, size, glucose homeostasis, and apoptosis. This provides a valuable understanding of the possible mechanisms involved in PanNETs [33].

Islet transplantation is a promising cell replacement therapy for restoring glycometabolic function in severe diabetic patients, yet the success of this approach is often hindered by the failure of many transplanted islets to engraft. New strategies are needed to enhance graft survival and improve clinical outcomes. We have highlighted the role of PHLDA3 in this context. While PHLDA3 is known as a suppressor of neuroendocrine tumorigenicity, its deficiency has been shown to increase islet proliferation, prevent apoptosis, and enhance insulin-releasing functions without leading to tumor formation. We found that *PHLDA3*-deficient islets showed promise in transplantation: when transplanted into diabetic mice, they conferred a significant improvement in glycometabolic control. This enhancement was primarily due to increased cell survival during the early stages of transplantation. This enhanced engraftment is associated with elevated AKT activity in *PHLDA3*-deficient islets, especially under hypoxic conditions. Consequently, *PHLDA3*-deficient islets demonstrate greater resistance to the stresses of islet isolation and transplantation. These findings suggest that targeting *PHLDA3* expression could be a novel and effective strategy for improving islet engraftment and glycemic control in diabetic patients undergoing islet transplantation [35].

### 9.2. PHLDA3 in PanNETs

Prior research has established a connection between loss of heterozygosity (LOH) at the 1q31 chromosomal region, where the *PHLDA3* gene resides, with PanNETs aggressiveness [45,46]. Loss of *PHLDA3* leads to increased AKT activation and subsequently increased signaling through the PI3K/AKT/mTOR pathway, which is commonly upregulated in PanNETs [47]. Recently, hyper-methylation of *p53* and its downstream target *PHLDA3* promoter regions has been reported, suggesting epigenetic and genetic regulation of *p53* signaling in PanNETs development [33,48,49]. We found that the loss of *PHLDA3* expression in PanNETs is caused by a combination of LOH and promoter methylation (Figure 5A) [33]. Analysis utilizing a microsatellite marker in close proximity to the *PHLDA3* gene revealed LOH in a significant 72% of human PanNETs samples. Furthermore, PanNETs specimens that exhibited LOH showed substantially reduced expression of PHLDA3 compared to specimens without LOH. Interestingly, PanNETs consistently exhibited methylation in the remaining *PHLDA3* allele transcriptional regulatory region. The *PHLDA3* gene in PanNETs is efficiently silenced through a two-hit inactivation mechanism, including LOH and methylation. The mechanism of downregulation by promoter methylation involves the addition of methyl groups to CpG islands within the promoter region of the PHLDA3 gene. This modification represses transcription by preventing the binding of transcription factors and by recruiting methyl-binding proteins that form a repressive chromatin structure. In PanNETs, a CpG island overlapping the promoter region and the first exon of PHLDA3 has been identified as a site of frequent methylation. Methylation at this site is tightly linked to transcriptional silencing, as demonstrated by significantly reduced PHLDA3 mRNA expression in PanNETs having promoter methylation. Treatment with the demethylating agent 5-aza-C led to a recovery of PHLDA3 expression, confirming that DNA methylation is a key mechanism regulating its silencing. These findings identify *PHLDA3* as a novel tumor suppressor gene with potential implications for PanNETs development [33]. Building on the identification of *PHLDA3* as a two-hit inactivated tumor suppressor in PanNETs, studies reported a connection between this gene and the tumor suppressor genes associated with multiple endocrine neoplasia type 1 (*MEN1*) in these tumors [50,51,52,53]. *MEN1* mutations and LOH are frequently observed in PanNETs. Interestingly, we found that LOH analysis demonstrated a comparable frequency of LOH for the *MEN1* gene (67%) compared to *PHLDA3* (72%) [33]. Furthermore, a significant proportion of PanNETs displayed LOH at both the *PHLDA3* and *MEN1* loci, suggesting these genes do not function in the same linear pathway (Figure 5B,C) [33]. In PanNETs, the frequency of LOH at the PHLDA3 locus is linked to a malignant phenotype, and patients with LOH at this locus have a poorer prognosis compared to those without LOH. Additionally, lower PHLDA3 expression is correlated with a poorer prognosis in patients with cervical SCC, as mentioned in the manuscript. Based on this, we expect low PHLDA3 expression to be associated with a poor survival rate.

### 9.3. PHLDA3 in Other NETs

A study discovered a significant prevalence of *PHLDA3* gene abnormalities in lung neuroendocrine tumors (LNETs), particularly in comparison to other forms of lung cancer [19]. Furthermore, LNETs exhibited decreased PHLDA3 expression and increased AKT activation compared to normal lung tissues. This correlation suggests that functional loss of PHLDA3 can contribute to AKT activation in LNET development in these specific cancers [5,54]. In addition, phosphatase and tensin homolog (PTEN) and PHLDA3 utilize distinct mechanisms to inhibit AKT signaling [5]. Interestingly, our analysis of large-cell neuroendocrine carcinoma (LCNEC) samples revealed that loss of *PTEN* and *PHLDA3* often occurs in the same tumors, and these losses have an additive effect on AKT activation [5]. We reported a connection between the loss of PHLDA3 and the functioning of *p53* in LNETs, as *PHLDA3* is a gene that is activated by *p53*. Remarkably, LNETs that have wild-type *p53* exhibited a substantial occurrence (63%) of PHLDA3 loss. In contrast, tumors with non-functional *p53* had a significantly lower frequency (13%) of PHLDA3 loss [19]. This substantial difference suggests that an intact *p53* pathway is crucial for maintaining PHLDA3 expression. In total, 91% of LNETs exhibited functional loss of either *p53* or *PHLDA3* [19]. This data strongly suggests that defects in this *p53*–PHLDA3 pathway play a major role in LNET development. There is an association between PHLDA3 expression levels and cancer patient survival.

Rectal NETs are often presented as tiny, easily missed lesions, particularly without lymph node involvement. In order to gain a deeper understanding of the factors that contribute to the development of tumors and their spread to other parts of the body, we conducted a study on 55 rectal NET specimens. We found that approximately 60% of the specimens showed LOH at the *PHLDA3* locus. Interestingly, a high frequency of LOH was also observed at the *MEN1* gene locus. Like PanNETs, LOH events at both *PHLDA3* and *MEN1* frequently co-occurred, indicating these genes function in distinct tumor-suppressing pathways [55]. Therefore, rectal NET development likely requires the functional loss of both pathways, mediated by PHLDA3 and MEN1.

## 10. *PHLDA3* as a Prognostic Marker and Therapeutic Target for Cancer

AKT plays a pivotal role in cancer cell migration, invasion, and angiogenesis, all essential for tumor growth and metastasis. It mediates Vascular Endothelial Growth Factor (VEGF)-induced endothelial cell migration through actin reorganization, with myristylated AKT enhancing this effect [56]. Additionally, endothelial nitric oxide synthase (eNOS) activation via AKT phosphorylation of Ser-1177 is vital for VEGF-mediated endothelial cell migration [57,58]. AKT also impacts EMT by regulating EMT-inducing transcription factors. Phosphorylated AKT downregulates GSK3β, leading to the stabilization and nuclear accumulation of β-catenin and Snail, which promotes EMT [59,60,61,62,63,64]. This mechanism is consistent with invasive cancers, where increased AKT phosphorylation downregulates GSK3β and results in Snail overexpression. Since PHLDA3 regulates AKT through inhibition of its signaling pathway, it can influence AKT-mediated processes in cancer progression. Therefore, PHLDA3 can be involved in metastatic regulation. Given its regulatory role, PHLDA3 can be used as a prognostic marker and therapeutic target in many cancers. Recent research underscores PHLDA3’s significant role in various cancers. In PanNETs, LOH at the *PHLDA3* locus correlates with higher malignancy and poorer prognosis, emphasizing *PHLDA3*’s critical role in suppressing tumor progression [33]. Reduced PHLDA3 expression in ESCC is associated with poor patient survival and increased risk of postoperative tumor progression and recurrence [44]. PHLDA3 also shows promise as a biomarker for HNSCC, with elevated expression correlating with better prognosis in patients with wild-type *p53*, while lower expression in *p53*-mutated SCCs is linked to poorer outcomes [11]. Similarly, higher PHLDA3 levels correlate with better prognosis in cervical SCCs, although this is not directly linked to *p53* status [10]. In prostate cancer, 22% of patients exhibit decreased PHLDA3 expression, with clinical recurrence associated with significant DNA methylation of the *PHLDA3* gene. In lung neuroendocrine tumors (NETs), PHLDA3 promoter methylation has been identified as a key mechanism responsible for its downregulation. Studies have shown that hypermethylation of CpG islands within the PHLDA3 promoter region leads to transcriptional silencing, contributing to tumor progression. Specifically, lung NET cell lines such as H1299 exhibit significant PHLDA3 methylation, resulting in low gene expression. Treatment with the DNA-demethylating agent 5-aza-2′-deoxycytidine restored PHLDA3 expression, confirming that promoter methylation directly represses its transcription [65]. In breast cancer, lower PHLDA3 mRNA levels are observed, with the most significant reductions in HER2 + subtypes and those with ER-negative and *TP53* mutations [66]. Additionally, decreased PHLDA3 expression is linked to poor prognosis in colorectal cancer and osteosarcoma patients, reflecting its potential role in *p53*-mediated tumor suppression [6,34]. Collectively, these results demonstrate that PHLDA3 plays a significant role in cancer progression and patient prognosis across various cancers. Its ability to regulate AKT signaling highlights its potential as a prognostic marker and therapeutic target, emphasizing its importance in tumor suppression and metastasis control.

## 11. Future Directions and Drug Development Challenges

Since PHLDA3 expression is regulated epigenetically, targeting DNA methylation or histone modifications could offer a viable therapeutic approach alongside strategies to stabilize the protein or enhance its interactions with key signaling molecules. However, translating these strategies into effective clinical therapies requires addressing issues such as off-target effects, bioavailability, and tumor heterogeneity. High-throughput drug screening could help identify small molecules that enhance PHLDA3 expression or function, paving the way for clinical studies to validate its role as a prognostic marker and therapeutic target. Large-scale patient studies and clinical trials investigating drugs that restore PHLDA3 expression or inhibit the PI3K/AKT pathway in PHLDA3-deficient tumors are essential for assessing its therapeutic potential. Additionally, integrating multi-omics approaches, such as genomics, transcriptomics, and proteomics, will aid in uncovering novel regulatory mechanisms and optimizing combination therapies. Co-targeting the PI3K/AKT pathway alongside strategies to boost PHLDA3 levels may yield synergistic effects, and exploring its role in immuno-oncology could provide new insights into tumor immune evasion and response to immunotherapy.

## 12. Conclusions

This review underscores the pivotal role of PHLDA3 in cancer progression, particularly in EMT, invasion, and metastasis. Loss of PHLDA3 expression is linked to poor prognosis across various cancers, including NETs and SCCs. PHLDA3’s tumor-suppressive function involves inhibiting the PI3K/AKT signaling pathway, which is crucial in EMT and cancer cell proliferation. Additionally, studies highlight PHLDA3’s role in modulating the Wnt/β-catenin pathway, further influencing EMT and metastasis. These findings position PHLDA3 as a promising prognostic marker and therapeutic target, with its regulation offering potential strategies for combating cancer invasiveness and metastasis.

## Figures and Tables

**Figure 1 cancers-17-01069-f001:**
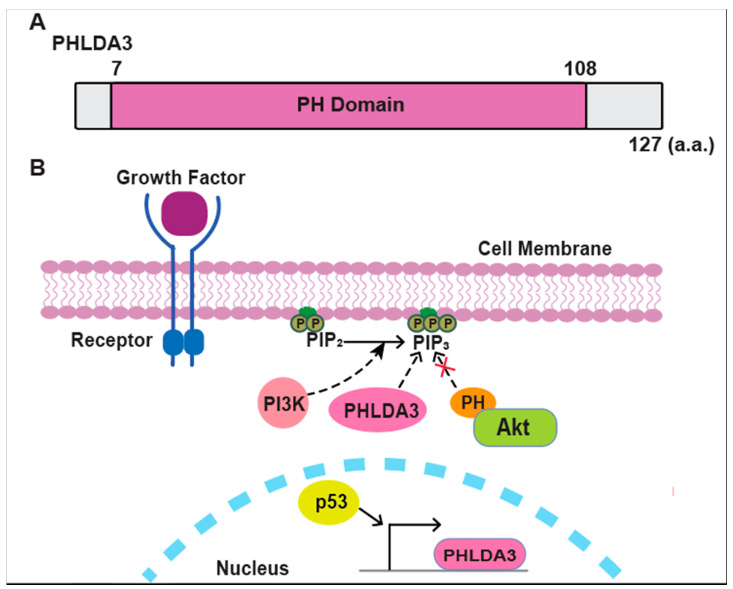
Role of PHLDA3 in the PI3K/AKT signaling pathway. (**A**) A schematic diagram of PHLDA3. PH–Pleckstrin Homology. (**B**) Growth factor stimulation activates PI3K, which converts PIP2 into PIP3, facilitating AKT recruitment to the cell membrane via its PH domain. PHLDA3 competitively binds to PIP3, thereby inhibiting AKT activation and downstream signaling. p53 binds to *PHLDA3* and regulates its expression at the transcriptional level.

**Figure 2 cancers-17-01069-f002:**
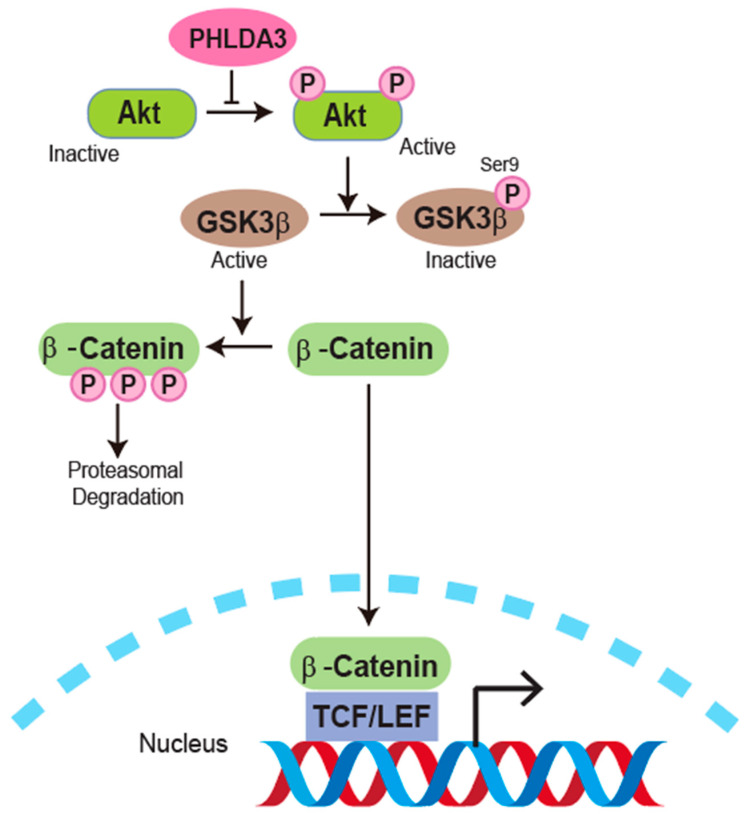
PHLDA3 and Wnt signaling pathway. PHLDA3 is shown to influence this pathway by regulating β-catenin stability and activity. In the absence of AKT activation, GSK3β remains active, facilitating the proteasomal degradation of β-catenin. When AKT becomes phosphorylated and activated, it inhibits GSK3β through phosphorylation at Ser9, preventing β-catenin degradation. Stabilized β-catenin translocates to the nucleus, where it interacts with T-cell factor/lymphoid enhancer factor (TCF/LEF) transcription factors to drive the expression of its target genes. PHLDA3 is depicted as a key regulator in this cascade, fine-tuning the balance between β-catenin degradation and activation.

**Figure 3 cancers-17-01069-f003:**
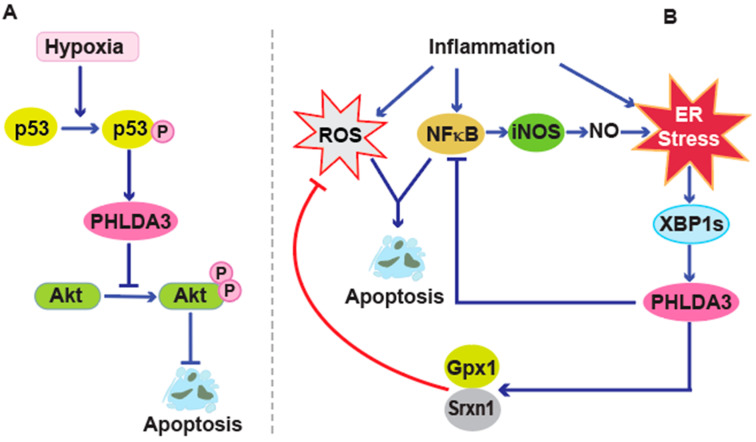
Dual function of PHLDA3 as both an apoptosis inducer and inhibitor. (**A**) Under hypoxic conditions, p53 is activated through phosphorylation. This activation of p53 subsequently induces the expression of PHLDA3. Once expressed, PHLDA3 inactivates AKT, leading to the induction of apoptosis. (**B**) Upon ER stress, XBP1 upregulates PHLDA3, which suppresses pro-inflammatory pathways by inhibiting the expression of NFκB and iNOS, reducing the production of inflammatory mediators like nitric oxide (NO). Additionally, PHLDA3 helps alleviate oxidative stress by promoting the mRNA expression of key antioxidant genes, such as Gpx1 and Srxn1, which regulate ROS levels. Through these mechanisms, PHLDA3 supports β cell survival by preventing apoptosis induced by inflammation and oxidative damage.

**Figure 4 cancers-17-01069-f004:**
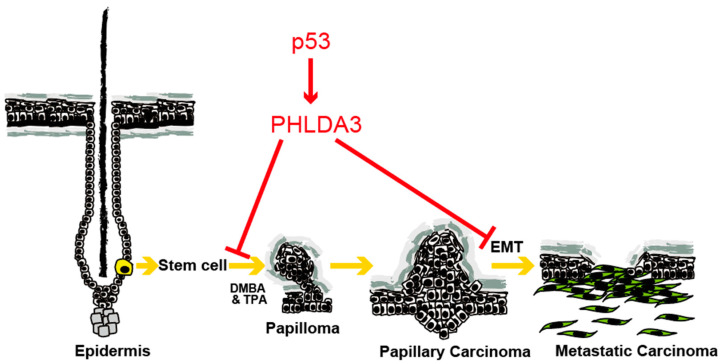
Role of PHLDA3 in SCC progression. PHLDA3 expression suppresses the development of papilloma and metastasis of SCC.

**Figure 5 cancers-17-01069-f005:**
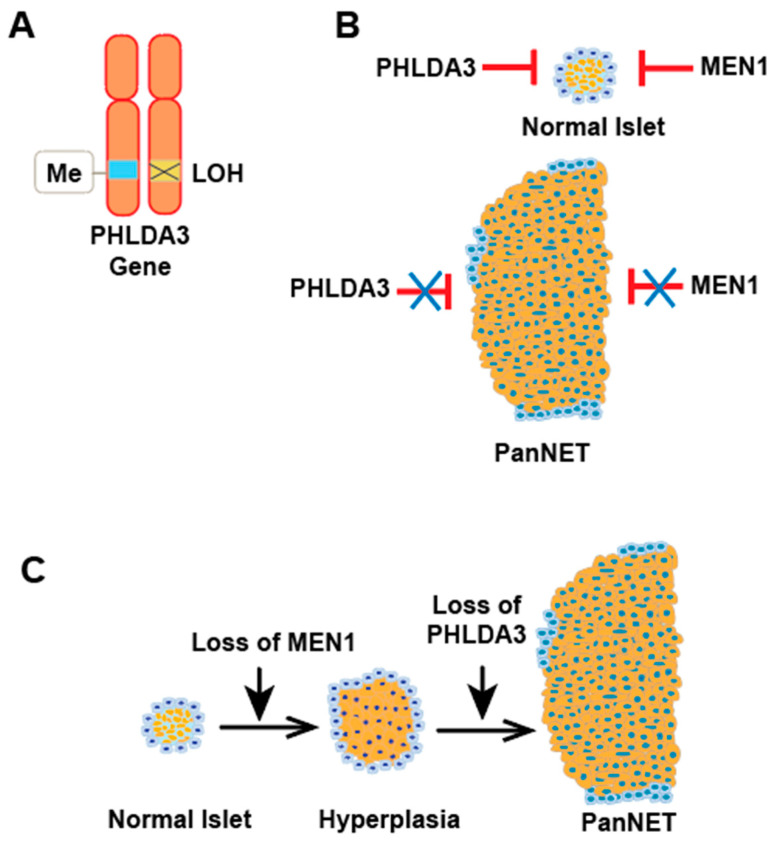
Defects in the PHLDA3 gene are observed in PanNETs. (**A**) PanNETs may experience two-hit inactivation of *PHLDA3* through LOH and methylation at its loci. (**B**) Tumorigenesis in PanNETs requires the loss of both PHLDA3 and MEN1 function, as these genes normally restrain cell proliferation in islet cells. (**C**) The progressive development of PanNETs involves the functional loss of PHLDA3 alongside MEN1, leading to islet abnormalities and eventual tumorigenesis.

**Table 1 cancers-17-01069-t001:** Summary of PHLDA3’s role in various cancer types and associated mechanisms.

Cancer Type	Key Findings	Reference
ESCC	PHLDA3 is downregulated in ESCC, leading to increased PI3K/AKT signaling, promoting proliferation, migration, invasion, and angiogenesis. Overexpression of PHLDA3 suppresses these effects.	[13]
OS	PHLDA3 inhibits proliferation, migration, and chemoresistance in OS. miR-19a-3p negatively regulates PHLDA3, emphasizing its potential as a therapeutic target.	[6]
AML	PHLDA3 interacts with CITED2 to regulate apoptosis. CITED2 knockdown increases PHLDA3 expression, inducing apoptosis in AML cells.	[42]
B-cell Lymphoma	PHLDA3 suppresses the B-cell receptor (BCR) signaling pathway by blocking AKT activation, reducing lymphoma cell proliferation and survival.	[43]
Prostate Cancer	PHLDA3 is epigenetically silenced through promoter methylation. Overexpression of PHLDA3, reducing proliferation, division rate, and EMT.	[4]
cSCC	Loss of PHLDA3 promotes EMT and metastasis. PHLDA3 deficiency complements p53 mutations, leading to aggressive SCC progression.	[10]
HNSCC	PHLDA3 functions as a negative feedback regulator of PI3K signaling. Increased PHLDA3 expression may be involved in PI3K dysregulation in HNSCC.	[11]
PanNETs	LOH and promoter methylation contribute to PHLDA3 silencing, leading to AKT hyperactivation and tumor progression.	[33]
LNETs	PHLDA3 loss is frequent in LNETs and is associated with increased AKT activation. It functions as a tumor suppressor in this context.	[19]

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
