# Peer review of "The Role of PHLDA3 in Cancer Progression and Its Potential as a Therapeutic Target"

_cancers, 2025, doi:10.3390/cancers17071069_

Round 1
Reviewer 1 Report
Comments and Suggestions for Authors
The authors revised the role of role of Pleckstrin homology-like domain family A, member 3 (PHLDA3), a p53-regulated tumor suppressor protein, in tumor progression, invasion, metastasis, and therapeutic opportunities.
The article is well-written and referenced and tackles an interesting area in tumor biology and therapeutic implications. This review is novel and of interest to cancer researchers. The following revisions will strengthen the manuscript.
The authors should comment/discuss how this field should be moving forward and difficulties in developing drugs that increase the levels/activity of PHLDA3. Studies in this field are pre-clinical and they should stress the need for future clinical studies. The discussion should be more in depth and how novel approaches such as the use of multi-omics, combination treatments, among others may propel the field forward.
It would be interesting to comment why is it in some cancers PHLDA3 works as an oncogene?
Minor comments:
The listed reviews regarding AKT in tumorigenesis are outdated. The authors need to replace them by more recent ones such as: AKT and the Hallmarks of Cancer. Cancer Res. 2024
Line 351: C should be replaced by Fig. 5C
Abbreviations should be introduced in figure legends.
Fig. 5B legend is missing
Author Response
Reviewer 1
First of all, we thank the reviewer very much for their careful consideration of our manuscript; we appreciate the reviewers’ helpful comments and believe that our manuscript has been significantly improved as a result. Our responses are outlined below. We have addressed all the feedback and highlighted our responses in red font color in the manuscript for clarity.
1- The authors should comment/discuss how this field should be moving forward and difficulties in developing drugs that increase the levels/activity of PHLDA3. Studies in this field are pre-clinical and they should stress the need for future clinical studies. The discussion should be more in depth and how novel approaches such as the use of multi-omics, combination treatments, among others may propel the field forward.
We appreciate the reviewer’s insightful comment. We have carefully incorporated their feedback in lines 536-551.
2- It would be interesting to comment why is it in some cancers PHLDA3 works as an oncogene?
We thank the reviewer for this valuable comment. In some cancers, such as lung adenocarcinoma, PHLDA3 overexpression has been linked to increased cell proliferation and invasion. This paradox may stem from context-dependent interactions with signaling pathways. We have added an explanation regarding how PHLDA3 may function as an oncogene in certain cancers in lines 172-177.
Minor comments:
3- The listed reviews regarding AKT in tumorigenesis are outdated. The authors need to replace them by more recent ones such as: AKT and the Hallmarks of Cancer. Cancer Res. 2024:
Thank you for your comments. We have replaced the outdated references with more recent ones, including AKT and the Hallmarks of Cancer (Cancer Res. 2024), as references 21-23.
4- Line 351: C should be replaced by Fig. 5C:
We have replaced this.
5- Abbreviations should be introduced in figure legends.
We appreciate the reviewers' suggestion. The abbreviations have already been introduced in the manuscript to ensure clarity in the figure legends, in accordance with other manuscripts published in Cancers journal.
6- Fig. 5B legend is missing:
We appreciate the reviewer’s feedback. The legend for Fig. 5B has been added in lines 459-462.
Reviewer 2 Report
Comments and Suggestions for Authors
An intriguing review of PHLDA3 is its value as a prognostic marker; moreover, its role in suppressing cancer progression underscores its potential as a promising therapeutic target. Some points should be noted as below to further improve the quality of this paper,
1) The title “PHLDA3 in Cancer: Insights into its Role in Progression, Metastasis, Invasion, and Therapeutic Potential” prompts the question of whether the term 'Progression' encompasses 'Metastasis and Invasion' or if these processes are considered separately? In other words, if “Progression” does not include “invasion and metastasis”, then exactly what does it represent? Not only the title, but the entire paper should focus on this issue (e.g. line 61-62).
Lin 418 “This review underscores the pivotal role of PHLDA3 in cancer progression, particularly in EMT, invasion, and metastasis”, I think such a description is reasonable.
2) Simple Summary: “Downregulation of PHLDA3 is associated with increased EMT markers”, it is prone to giving rise to ambiguity, for example, E-cadherin is a marker of EMT, so “Downregulation of PHLDA3 is associated with increased E-cad”?
3) 1. Introduction “Cancer arises from uncontrolled cell division and infiltration of surrounding tissues, frequently resulting from genetic mutations in particular genes”, perhaps we have a more updated and comprehensive comprehension and cognition of cancer. In fact, cancer is proposed to be a pathological ecosystem that cancer cells dynamically interplay with their tumor microenvironment such as microorganisms and immune cell, and even co-evolve in a cross-level manner.( https://pubmed.ncbi.nlm.nih.gov/37056571/). So, in Introduction section, such view should be better updated.
4) Line “..including various cellular regulatory pathways, such as the Protein Kinase B (AKT) signaling pathway”, so many pathways, why is AKT, not other pathways?
5) “7. PHLDA3 in ESCC, Osteosarcoma, Acute Myeloid Leukemia, B-cell Lymphoma, and Prostate Cancer Cell Lines”, the contents of “7.PHLDA3 in ESCC” can be integrated into “8. The Role of PHLDA3 in SCC Tumor Progression and Metastasis”?
6) Since decreased PHLDA3 expression can promote EMT in SCCs, so how about the expression of PHLDA3 and EMT markers in cancer tissue, especially spindle cells in tissues? Because a considerable number of studies have indicated that spindle-shaped cancer cells are the morphological markers of EMT in tumors (https://pubmed.ncbi.nlm.nih.gov/25420898/; https://pubmed.ncbi.nlm.nih.gov/24349446/; https://pubmed.ncbi.nlm.nih.gov/22486228/; https://pubmed.ncbi.nlm.nih.gov/19381684/).
It should be better discussed (Not mandatory)
Author Response
Reviewer 2
First of all, we thank the reviewer very much for their careful consideration of our manuscript; we appreciate the reviewers’ helpful comments and believe that our manuscript has been significantly improved as a result. Our responses are outlined below. We have addressed all the feedback and highlighted our responses in red font color in the manuscript for clarity.
1) The title “PHLDA3 in Cancer: Insights into its Role in Progression, Metastasis, Invasion, and Therapeutic Potential” prompts the question of whether the term 'Progression' encompasses 'Metastasis and Invasion' or if these processes are considered separately? In other words, if “Progression” does not include “invasion and metastasis”, then exactly what does it represent? Not only the title, but the entire paper should focus on this issue (e.g. line 61-62).
Line 418 “This review underscores the pivotal role of PHLDA3 in cancer progression, particularly in EMT, invasion, and metastasis”, I think such a description is reasonable.
We agree with this point and have made the necessary changes accordingly. The distinction between 'Progression' and 'Metastasis/Invasion' is important, and I have revised the manuscript to ensure clarity on this issue throughout. Additionally, I have reviewed and updated the relevant sections in lines 3, 15, 33, and 71 to reflect these changes
2) Simple Summary: “Downregulation of PHLDA3 is associated with increased EMT markers”, it is prone to giving rise to ambiguity, for example, E-cadherin is a marker of EMT, so “Downregulation of PHLDA3 is associated with increased E-cad”?
Thank you for your comment. To avoid ambiguity, we have clarified that 'Downregulation of PHLDA3 is associated with increased expression of mesenchymal EMT markers,' which is a characteristic function in SCCs, in lines 17-18.
3) 1. Introduction “Cancer arises from uncontrolled cell division and infiltration of surrounding tissues, frequently resulting from genetic mutations in particular genes”, perhaps we have a more updated and comprehensive comprehension and cognition of cancer. In fact, cancer is proposed to be a pathological ecosystem that cancer cells dynamically interplay with their tumor microenvironment such as microorganisms and immune cell, and even co-evolve in a cross-level manner (https://pubmed.ncbi.nlm.nih.gov/37056571/). So, in Introduction section, such view should be better updated.
We thank the reviewer for this valuable comment and have added a more detailed explanation of cancer, highlighting its uncontrolled cell growth, metastatic nature, and influences from external and internal factors (lines 39-44).
4) Line “.including various cellular regulatory pathways, such as the Protein Kinase B (AKT) signaling pathway”, so many pathways, why is AKT, not other pathways?
Thank you for your comment. We have added the Wnt signaling pathway to our description alongside AKT in line 50, as these two pathways are the main focus of this review. Our discussion emphasizes their roles in PHLDA3 regulation and their impact on cancer progression.
5) “7. PHLDA3 in ESCC, Osteosarcoma, Acute Myeloid Leukemia, B-cell Lymphoma, and Prostate Cancer Cell Lines”, the contents of “7. PHLDA3 in ESCC” can be integrated into “8. The Role of PHLDA3 in SCC Tumor Progression and Metastasis”?
Thank you so much. We have changed the title of Section 8 to PHLDA3 in SCC: Functional Roles in Tumor Progression and Therapeutic Resistance in line 328. This title highlights the specific characteristics of SCC tumors, particularly their progression, including metastatic behavior, and resistance mechanisms, distinguishing it from the general discussion of PHLDA3 in various cancer cell lines in Section 7.
6) Since decreased PHLDA3 expression can promote EMT in SCCs, so how about the expression of PHLDA3 and EMT markers in cancer tissue, especially spindle cells in tissues? Because a considerable number of studies have indicated that spindle-shaped cancer cells are the morphological markers of EMT in tumors (https://pubmed.ncbi.nlm.nih.gov/25420898/; https://pubmed.ncbi.nlm.nih.gov/24349446/; https://pubmed.ncbi.nlm.nih.gov/22486228/; https://pubmed.ncbi.nlm.nih.gov/19381684/).
It should be better discussed (Not mandatory).
Thank you for your valuable comment. While numerous studies have identified spindle-shaped cancer cells as morphological markers of EMT, to the best of our knowledge, there are no available studies specifically investigating PHLDA3 expression in spindle cells within tissues. Our current discussion is based on the established role of PHLDA3 in regulating EMT markers such as vimentin and N-cadherin in SCCs. Further studies are needed to explore the relationship between PHLDA3 expression and spindle cell morphology in tumor tissues.
Reviewer 3 Report
Comments and Suggestions for Authors
The paper of Kamel et al. is devoted to review the role of Pleckstrin homology-like domain family A member 3 (PHLDA3) in neoplasia. This is a good and comprehensive review covering different biological roles of PHLDA3 in different types of tumors.
The review is clearly written and is easy to understand.
Minor issues:
1) Figure 3, Legend. “mRNA expression” is written in red
2) I suggest authors to add the Table summarizing key publications on the role of PHLDA3 in different neoplasia
3) Is there any association between the level of PHLDA3 expression and the survival rate of cancer patients?
4) What other transcriptional factors beyond p53 that up-regulates PHLDA3?
Author Response
Reviewer 3
First of all, we thank the reviewer very much for their careful consideration of our manuscript; we appreciate the reviewers’ helpful comments and believe that our manuscript has been significantly improved as a result. Our responses are outlined below. We have addressed all the feedback and highlighted our responses in red font color in the manuscript for clarity.
Minor issues:
1) Figure 3, Legend. “mRNA expression” is written in red,
Thank you for your suggestion. We have revised Figure 3 and changed the "mRNA expression" text color from red to black in the legend as requested.
2) I suggest authors to add the Table summarizing key publications on the role of PHLDA3 in different neoplasia:
Thank you for your suggestion. I have added a table summarizing key publications on the role of PHLDA3 in different neoplasias in lines 533-534.
3) Is there any association between the level of PHLDA3 expression and the survival rate of cancer patients?
Thank you for your question. We have added an explanation regarding the association between PHLDA3 expression levels and cancer patient survival in lines 434-439."
4) What other transcriptional factors beyond p53 that up-regulates PHLDA3?
We have already discussed some of these transcription factors in the manuscript, and we appreciate your suggestion. However, it is important to note that BARX2 and XBP1 are not direct transcription factors for PHLDA3 in the same manner as p53. We added this explanation in lines 157 and 243. Nevertheless, their regulatory effects contribute to the modulation of PHLDA3 in specific cellular contexts.
BARX2 (BarH-like homeobox 2)
Role: Activates PHLDA3 transcription, leading to suppression of PI3K/AKT signaling in esophageal squamous cell carcinoma (ESCC).
Reference: Chen et al., Exp. Cell Res., 2023
XBP1 (X-box binding protein 1)
Role: Upregulates PHLDA3 expression during endoplasmic reticulum (ER) stress, helping regulate cell survival and stress responses.
Reference: Bensellam et al., Sci. Rep., 2019
Reviewer 4 Report
Comments and Suggestions for Authors
PHLDA3 is a -downstream effector of p53, and its expression level is regulated by this, cancer development guarding, transcription factor. Accordingly, PHLDA3 plays key roles in suppressing or propelling malignant transformation, by negating or potentiating the effects of major oncogenic pathways like PI3K/Akt , and Wnt/b-Catenin, respectively. Consequently, PHLDA3 governs basic oncogenic processes like cell proliferation, and epithelial-mesenchymal transition (EMT) , that drives metastatic dissemination and therapy resistance acquirement, of malignant diseases. The intricate roles of PHLDA3 in different cancer types, and their potential translational ramifications, merits a summarizing review of this interesting protein. Indeed, this review by Kamel et al. is comprehensive, thorough, and widens our knowledge and understanding of the suppressive or promoting activities of PHLDA3, in cancer progression. Concomitantly, the potential implications of the PHLDA3 regulatory features, on cancer prognosis/diagnosis and intervention, is also being discussed.
Two points should be addressed before this review is being accepted for publication:
- The authors should add a descriptive figure, depicting the structure of PHLDA3, , highlighting the various functional domains of the protein, and marking their roles like mediation of regulatory interactions with basic oncogenic factors.
- Based on the regulatory and expression patterns of PHLDA3, the authors propose the adoption of this protein as a prognostic/diagnostic tools, and as a potential target for cancer intervention. In this later aspect the authors should also elaborate on the putative druggability of a protein like PHLDA3.
Author Response
Reviewer 4
First of all, we thank the reviewer very much for their careful consideration of our manuscript; we appreciate the reviewers’ helpful comments and believe that our manuscript has been significantly improved as a result. Our responses are outlined below. We have addressed all the feedback and highlighted our responses in red font color in the manuscript for clarity.
Two points should be addressed before this review is being accepted for publication:
1) The authors should add a descriptive figure, depicting the structure of PHLDA3, , highlighting the various functional domains of the protein, and marking their roles like mediation of regulatory interactions with basic oncogenic factors.
Thank you for your comments. We have added a descriptive figure depicting the structure of PHLDA3, highlighting its various functional domains and their roles in mediating regulatory interactions with key oncogenic factors, as requested. This has been included in line 113, and the relevant content has been modified in lines 89-92.
2) Based on the regulatory and expression patterns of PHLDA3, the authors propose the adoption of this protein as a prognostic/diagnostic tools, and as a potential target for cancer intervention. In this later aspect the authors should also elaborate on the putative druggability of a protein like PHLDA3.
Thank you for your insightful comment. We have included a paragraph regarding the Future Directions and Drug Development Challenges in lines 536-551.
Reviewer 5 Report
Comments and Suggestions for Authors
In this manuscript titled "PHLDA3 in Cancer: Insights into its Role in Progression, Metastasis, Invasion, and Therapeutic Potential", the authors review the scientific literature regarding PHLDA3, a recently characterized tumor suppressor in some solid tumors.
Since this is a biomarker whose functional activities in cancer are recently identified, the literature counts only 45 papers of which 3 are reviews not specifically focused on PHLDA3 (PUBMED, key words: PHLDA3, cancer, tumor). Therefore, the authors have well hit a niche in which to place this manuscript, with a very direct and specific topic.
Overall the review is comprehensive of the state of the art, the following requests should be supported:
-Paragraph 4: Please explain the Wnt signaling pathway better, as the authors mention beta catenin after without having described it in the pathway.
-Explain better the mechanism of downregulation by promoter methylation and in which tumors it has been seen.
The English style should be revised throughout the manuscript. Some suggestions at improving sentences:
-Line 43: “…and suppresses tumors “ , resulting in tumor suppression.
-Line 72: "promote survival", control cell cycle progression. p53 pathway also partly controls the transcription of apoptotic genes along with those of cell cycle arrest, autophagy, and senescence.
-Line 153: "A study by Bensellam et al....". Describe the model of this study further, mention that it deals with pancreas.
Comments on the Quality of English LanguageThe English style should be revised throughout the manuscript. Some suggestions at improving sentences:
-Line 43: “…and suppresses tumors “ , resulting in tumor suppression.
-Line 72: "promote survival", control cell cycle progression. p53 pathway also partly controls the transcription of apoptotic genes along with those of cell cycle arrest, autophagy, and senescence.
Author Response
Reviewer 5
First of all, we thank the reviewer very much for their careful consideration of our manuscript; we appreciate the reviewers’ helpful comments and believe that our manuscript has been significantly improved as a result. Our responses are outlined below. We have addressed all the feedback and highlighted our responses in red font color in the manuscript for clarity.
Comments and Suggestions for Authors
In this manuscript titled "PHLDA3 in Cancer: Insights into its Role in Progression, Metastasis, Invasion, and Therapeutic Potential", the authors review the scientific literature regarding PHLDA3, a recently characterized tumor suppressor in some solid tumors.
Since this is a biomarker whose functional activities in cancer are recently identified, the literature counts only 45 papers of which 3 are reviews not specifically focused on PHLDA3 (PUBMED, key words: PHLDA3, cancer, tumor). Therefore, the authors have well hit a niche in which to place this manuscript, with a very direct and specific topic.
Overall the review is comprehensive of the state of the art, the following requests should be supported:
1- Paragraph 4: Please explain the Wnt signaling pathway better, as the authors mention beta catenin after without having described it in the pathway.
Thank you for your comment. I have now provided a clearer explanation of the Wnt signaling pathway, including its key components and role, before introducing β-catenin to ensure better clarity in lines 140-149.
2- Explain better the mechanism of downregulation by promoter methylation and in which tumors it has been seen.
Thank you for your comments. We have now included a more thorough explanation of the mechanism of downregulation by promoter methylation and its occurrence in various tumors, as requested. This has been added in lines 415-425 and 516-523.
-Line 153: "A study by Bensellam et al....". Describe the model of this study further, mention that it deals with pancreas.
Thank you for your question. We have now included a more thorough explanation of the model used in the study by Bensellam et al., specifically mentioning that it deals with pancreatic cells, in lines 232-241.
3- Comments on the Quality of English Language
The English style should be revised throughout the manuscript. Some suggestions at improving sentences:
4-Line 43: “…and suppresses tumors “ , resulting in tumor suppression.
Thank you for your comment. We have changed the phrase to 'suppresses tumors' for clarity to suppress tumors in line 52.
5-Line 72: "promote survival", control cell cycle progression. p53 pathway also partly controls the transcription of apoptotic genes along with those of cell cycle arrest, autophagy, and senescence.
Thank you for your comment, we added this in lines 80-82.
Round 2
Reviewer 1 Report
Comments and Suggestions for Authors
The authors have adequately answered my revisions and concerns.
Reviewer 2 Report
Comments and Suggestions for Authors
The author has made excellent revisions to most of the comments. Introducing such a novel perspective ( cancer is a complex ecosystem that cancer cells dynamically interplay with their tumor microenvironment such as microorganisms and immune cell, and even co-evolve in a cross-level manner (https://pubmed.ncbi.nlm.nih.gov/37056571/). ) I believe it will offer a different experience for the readers, but which is up to you to decide.
Reviewer 5 Report
Comments and Suggestions for Authors
The manuscript has been sufficiently improved. The latest version is acceptable for publication.